# Model-aware Counterfactual Data based Contrastive Decoding for Video-LLM

## Abstract

Video language models (Video-LLMs) are prone to hallucinations, often generating plausible but ungrounded content when visual evidence is weak, ambiguous, or biased. Existing decoding methods, such as contrastive decoding (CD), rely on random perturbations to construct contrastive data for mitigating hallucination patterns. However, such a way is hard to control the visual cues that drive hallucination or well align with model weaknesses. We propose Model-aware Counterfactual Data based Contrastive Decoding (GeWu), a new inference strategy that combines model-guided counterfactual construction with decoding. Our approach uses the Video-LLM's own feedback to identify object regions most responsible for hallucination, generating targeted counterfactual inputs at the object level rather than arbitrary frame or temporal modifications. These model-aware counterfactual data is then integrated into CD to enforce evidence-grounded token selection during decoding. Experiments on EventHallusion, MVBench, and Perception-test show that GeWu consistently reduces hallucination while maintaining or improving task accuracy across diverse Video-LLMs, including InternVL3, Qwen2.5-VL and Qwen2-VL families. The method is especially effective in challenging scenarios involving small, occluded, or co-occurring objects. Our code and data will be publicly released.

## 1 Introduction

Video large language models (Video-LLMs) (Zhang et al., 2024b) have recently achieved remarkable progress, showing strong performance on tasks such as video question answering, reasoning, and captioning (Yuan et al., 2025). However, despite these advances, Video-LLMs are prone to hallucination, generating content that is plausible but not grounded in the actual visual evidence (Cai et al., 2025). This issue often arises when visual cues are weak or ambiguous, and is exacerbated by object-level uncertainty and dataset biases (Zhao et al., 2025). Such hallucinations undermine the reliability of Video-LLM outputs, reducing factual accuracy, distorting semantic understanding, and ultimately limiting their practical utility in real-world applications (Suo et al., 2025).

To mitigate hallucination, contrastive decoding (CD) has been introduced as an effective solution (Lee & Song, 2025). CD works by formulating generation as a contrastive search objective, ranking candidate tokens based on the difference between their log-likelihood under two different views (commonly the original data and its perturbed counterpart). Then, CD retains only those tokens with relatively higher plausibility from the original data than the perturbed view (Li et al., 2022), which leverages the contrastive view to identify the correct tokens without training. Crucially, the success of CD depends on the quality of the auxiliary perturbed data. If it accurately hits the model's weaknesses, CD can effectively reduce hallucination.

Most existing approaches rely on creating perturbed data pairs through random perturbation (*e.g.,* add Gaussian noise and frame dropping). Because these modifications are not tied to what the model is unaware or relevant with the query, they can not guarantee alignment with the visual factors responsible for hallucination. As a result, the perturbed new data might fail to align with the model's actual weaknesses, leading to unstable or even degraded performance.

To solve this issue, this paper introduces a model-aware pipeline to construct high-quality counterfactual data for contrastive decoding in video-LLMs. A key feature of the method is that the

counterfactual data is built based on the feedback from the Video-LLM itself. Concretely, we compute the gradients of the Video-LLM, to adjust the perturbation strategy for maximizing the query reconstruction loss. In this way, the perturbation strategy will be optimized to construct a new video where the video-LLM required key information has been wiped, which directly reflects the Video-LLM's weaknesses. Thus, by contrasting with the counterfactual sample, the video-LLM will be encouraged to assign relatively higher plausibility for the tokens that are relevant to its required key information (*e.g.,* key objects and key frames), leading to less hallucination and higher accuracy.

To this end, we propose an inference-time intervention method for video-LLMs, namely GeWu, denoting Model-aware Counterfactual data based Contrastive Decoding. In our approach, given a video and the input query, we first identify all the objects through a detector (*e.g.,* YOLOv11), and assign soft masks to each object and each frame. Then, the masked video is fed to the Video-LLM to compute next token prediction loss for the input query, and the computed gradients are used to update the soft masks towards maximizing that loss. After multi-step optimization, we get the counterfactual video and feed it with the original video to the video-LLM, and perform contrastive decoding to obtain the final answer. Extensive experiments on EventHallusion(Zhang et al., 2024a), MVBench (Li et al., 2024), and perception-test (Patraucean et al., 2023) have shown the superiority of our method than existing contrastive decoding strategies. Furthermore, the consistent improvements on InternVL3-8B (Zhu et al., 2025), Qwen2.5-VL-7B/3B (Bai et al., 2023), Qwen2-VL-7B/2B (Wang et al., 2024) also demonstrate the effectiveness of our approach in different backbone models.

This work makes three primary contributions:

- We introduce a model-aware data augmentation strategy that leverages the Video-LLM's own feedback to construct counterfactual video data.

- We utilize the above strategy to compose object-level and frame-level counterfactual data, which can greatly improve the contrastive decoding method.

- Extensive experiments show that the proposed method can reduce hallucination and improve task accuracy across diverse benchmarks and models.

## 2 RELATED WORK

We review related work from the following three aspects.

**Vision-Language Models.**  Vision-language models (Video-LLMs) have rapidly advanced by coupling powerful visual encoders with large language models (LLMs). Early systems such as BLIP-2 (Li et al., 2023a) and Flamingo (Alayrac et al., 2022) introduced query-based transformers and gated cross-attention to bridge the modality gap, while the LLaVA (Lin et al., 2023) showed that an instruction-tuned LLM with a projection layer can effectively adapt visual features, enabling versatile agents like MiniGPT-4 (Zhu et al., 2023) and InstructBLIP (Dai et al., 2023). This paradigm has been extended to the temporal domain with Video-LLMs such as Video-LLaMA (Zhang et al., 2023) and Video-ChatGPT (Maaz et al., 2023), and further scaled in recent open-source systems (Zheng et al., 2024). Although these systems perform well on many video understanding tasks, recent analyses show that Video-LLMs still suffer from object and event hallucinations, especially in long videos with cluttered scenes and complex temporal relations (Huang et al., 2025; Ji et al., 2023). These limitations motivate dedicated work on measuring and mitigating hallucinations in multimodal models, which we review next.

**Hallucination in LLMs.**  Hallucination, the generation of plausible but factually incorrect content, is a central challenge for large language and vision-language models (Huang et al., 2025). Its causes include spurious dataset correlations, parametric knowledge gaps, and failures to ground predictions in the input evidence (Ji et al., 2023). In vision-language settings, hallucination can appear as object hallucination (describing non-existent entities), attribute hallucination (misstating properties), or relation hallucination (incorrect spatial or functional relations) (Rohrbach et al., 2018). To evaluate such errors, benchmarks such as POPE (Li et al., 2023b) and MM-HallBench (Geng et al., 2024) have been proposed, along with metrics like CHAIR (Rohrbach et al., 2018) that quantify fine-grained object hallucinations.

Existing work tackles hallucination from several angles. Training-time methods reshape the model's behavior by finetuning on curated negative data (Lin et al., 2021) or using reinforcement learning to penalize hallucinations, as in LURE (Zhou et al., 2023). Post-hoc approaches edit or rerank generated outputs with an external verifier, such as Woodpecker (Yin et al., 2024). A third line intervenes at inference time without changing model parameters. This family includes contrastive decoding (CD) methods and other sampling-based strategies that use additional views or signals to steer generation: for example, self-consistency decoding aggregates multiple samples to reject unstable answers (Wang et al., 2022; Taubenfeld et al., 2025), while self-reflection and self-refinement methods let the model critique and revise its own outputs (Madaan et al., 2023; Kim et al., 2023). Recent multimodal works such as VCD and DAMRO (Leng et al., 2024; Gong et al., 2024) instantiate this idea for vision-language models by leveraging contrastive visual views or internal representations. Our approach falls into this inference-time category, but focuses on constructing a model-aware counterfactual video view tailored to each query.

**Contrastive Decoding.** Contrastive decoding (CD) is a powerful training-free paradigm for improving faithfulness (Su & Xu, 2022). Its core idea is to steer generation by contrasting a strong "expert" distribution with a weaker "amateur" one. This principle has led to many variants that differ mainly in how the two distributions are defined. Some methods contrast internal logits from later versus earlier layers of a single model, as in DoLa (Mesgar et al., 2023), or exploit refined representation geometries, as in CAD (Mesgar et al., 2023). Others adapt CD to multimodal grounding by comparing distributions conditioned on different visual inputs or feature sets (Suo et al., 2025; Lee & Song, 2025). CD is also complementary to sampling-based strategies, where multiple candidate views (e.g., diverse decoding paths or self-reflective revisions) are scored and contrasted to select a more reliable answer (Cheng et al., 2024).

In multimodal settings, VCD (Leng et al., 2024) constructs a weaker view by applying heuristic input degradations such as occlusions or patch masking to the image or video, and then contrasts the outputs from the original and degraded inputs. DAMRO (Gong et al., 2024) operates internally, identifying visual tokens with low agreement between attention heads as hallucination-prone and suppressing them during decoding, while TruthX (Zhang et al., 2024c) prunes text spans that are not grounded in visual evidence using a verifier-like signal. These methods illustrate diverse ways of defining the weaker view or contrastive signal, ranging from fixed input perturbations to heuristics over internal states or external verifiers.

Our work follows the standard CD scoring rule but focuses on a different design choice: *how* to construct the negative view for Video-LLMs. Instead of using fixed heuristic perturbations or purely internal signals, we explicitly parameterize an object-level and frame-level mask over the input video and optimize its strengths with gradients from the same Video-LLM, conditioned on the query. This yields a model-aware counterfactual video that erases query-relevant evidence and serves as a maximally challenging contrastive view for the base video. In this way, GeWu instantiates CD in the input space of Video-LLMs through a learnable, gradient-guided counterfactual mask, while remaining training-free and compatible with existing decoding strategies.

## 3 PRELIMINARY

**Video Large Language Models.** A video large language model (Video-LLM) takes a video $V = (x_{1:T})$ and a text query $q$ as input, and produces a sequence of tokens as the response to the query $y_{1:m}$. In each step, the token $y_t$ is sampled based on the token probability distribution produced by the video-LLM. Formally, the above process can be formatted as:

$$p_\theta(y_{1:m} \mid V, q) = \prod_{n=1}^{m} p_\theta(y_n \mid y_{<n}, V, q).$$

(1)

Video-LLMs are prone to hallucination, generating content that is plausible but not grounded in the input video.

**Contrastive Decoding.** Contrastive decoding (CD) compares a base distribution with a reference distribution to sample the next token. It is widely used to reduce hallucination in LLMs. At step $n$, the probability for candidate token $t$ is

$$S_n(t) = \log p_\theta(t \mid V, q, y_{<n}) - \lambda \log p_\phi(t \mid V_{\text{ref}}, q, y_{<n}).$$

(2)

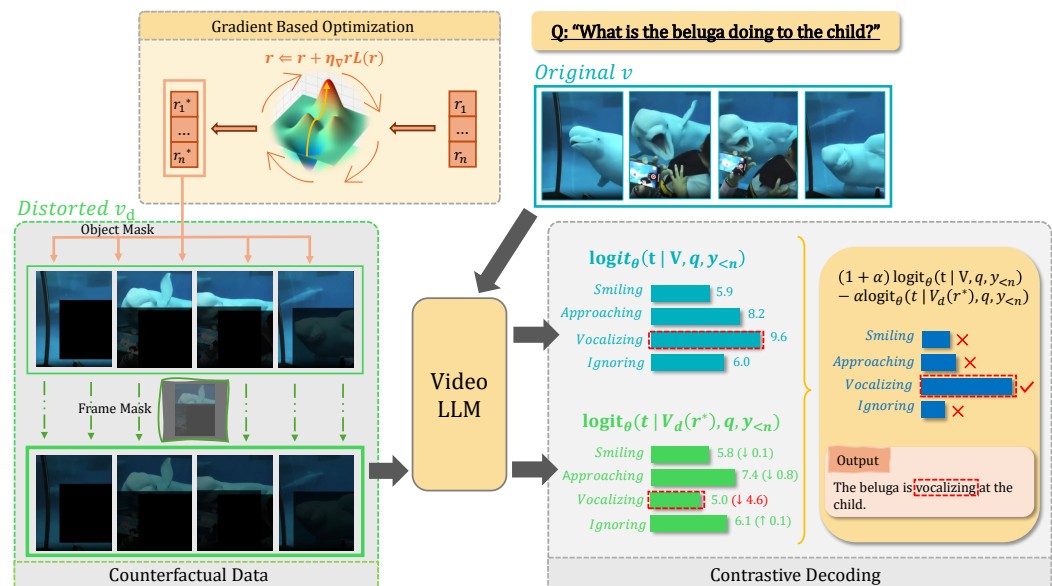

Figure 1: The overview of our proposed method. It consists of the model-aware counterfactual data augmentation using object-level and frame-level masks that can be optimized via the gradients from the VLM, and the contrastive decoding strategy to boost the probability of important tokens relevant to the masked part in the video.

Here $p_\theta$ and $p_\phi$ are produced by the Video-LLM conditioned on the original and perturbed video data, respectively. $V_{\text{ref}}$ is generally a noised or masked video. In this way, the tokens supported by the masked part will be down-weighted in $p_\phi$, leading to the probability increasing in $S_n(t)$.

**Our Target.** A limitation of prior work is that $V_{\text{ref}}$ is often created via random perturbations (*e.g.,* random frame dropping or masking), which can not well capture the visual cues that trigger hallucination. Our goal is to build *model–aware counterfactual inputs*: instead of random noise, the Video-LLM's own feedback determines which objects and which frames to mask and by how much. The result is a set of *spatial and temporal perturbed inputs* that highlight where hallucination arises. Combined with CD, the counterfactual data can well guide the video-LLM to suppress hallucinated mentions while retaining fluent, accurate responses.

# 4 METHOD

Our approach introduces a model-aware contrastive decoding pipeline designed to reduce object-level hallucinations in video-LLMs. The method operates in three stages. First, we construct counterfactual inputs by applying model-aware masks to video frames, producing controlled perturbations that preserve temporal consistency. Next, we optimize these masked views using the model's own feedback: regions that increase task loss when removed are identified as critical, ensuring that the augmented data directly reflects the model's weaknesses. Finally, these optimized counterfactual data is integrated into a contrastive decoding scheme, where tokens supported by grounded evidence are promoted and spurious mentions are suppressed. Note that our approach is training-free, in which the video-LLM remains frozen. The overview of our approach is shown in Figure 1.

## 4.1 COUNTERFACTUAL DATA AUGMENTATION

The first step of our approach is to construct object-aware masked data that serve as counterfactual inputs. Given an input video and a query, we first detect objects and track them across frames, then add masks in them. Pixels outside the masks remain unchanged. This procedure generates

the counterfactual data and initialize the mask for supporting following optimization strategy and contrastive decoding.

### 4.1.1 OBJECT DETECTION

Object detection is applied to each video frame to identify potential entities, with bounding boxes, class labels, and confidence scores produced for every detection. These per-frame results are then linked across time using heuristics that combine intersection-over-union overlap with motion cues, yielding stable object tracks rather than isolated detections. Each track is refined into a soft mask with smoothed boundaries, ensuring suitability for gradient-based updates, while temporal smoothing bridges short occlusions and fills gaps from missed detections. Overlapping objects are normalized by confidence so their masks share pixel space without exceeding full coverage, and unstable or low-quality tracks are removed. Importantly, small, occluded, or co-occurring objects are retained, as these are frequent sources of hallucination. The final output is a compact set of temporally consistent, interpretable object masks that serve as reliable evidence for constructing counterfactual data. Formally, let the input video be $V = \{x_t\}_{t=1}^T$ and the query be $q$. A YOLOv11-based detector produces an object set $\mathcal{O} = \{o_k\}_{k=1}^K$, each associated with per-frame masks $m_k^t \in [0, 1]^{H \times W}$.

### 4.1.2 MASKED DATA COLLECTION

We convert detected objects into a *masked view* that provides the perturbed visual inputs for counterfactual data construction. Each object $o_k$ in frame $x_t$ is associated with a soft mask $m_k^o \in [0, 1]^{H \times W}$ and a mask strength $r_k$. Masks from different objects are merged per pixel with a simple union mask

$$M^o = \max_k (\hat{r}_k \, m_k^p). \tag{3}$$

If several objects cover the same pixel, the one with the larger strength dominates that location. In addition to the spatial-level object mask, we also consider the temporal-level frame mask that perturbs few frame within the video, denoted as $M^f$. The final masked video is produced by adding the two kinds of masks into the original frames as:

$$V_d(\mathbf{r}) = V + r * (M^f + M^o), \tag{4}$$

To ensure stable gradient updates in Section 4.2, $r$ is used to control the strength of the mask for adjusting the counterfactual video data.

### 4.2 MODEL-AWARE CONTRASTIVE DATA OPTIMIZATION

The second stage leverages the model's own feedback to refine the masking strengths $\mathbf{r}$ in the counterfactual data. The intuition is straightforward: if obscuring a region increases task difficulty, then that region provides critical evidence. We therefore optimize $\mathbf{r}$ to maximize the loss of the Video-LLM to reconstruct the text input. This connects gradients directly to the data and creates a lightweight adversarial process in which inputs are perturbed while model parameters remain fixed.

Formally, given the query $q$, the prediction loss of its $n$-th token is denoted as:

$$\ell(q_n) = -\log p_\theta(q_n \mid V_d(\mathbf{r}), q_{<n}), \tag{5}$$

where $V_d(\mathbf{r}) = V + Z(\mathbf{r})$. Then, the strength $\mathbf{r}$ is updated by gradient ascent to maximize the loss:

$$\mathbf{r} \leftarrow \mathbf{r} + \eta \, \nabla_\mathbf{r} \, \frac{1}{N} \sum_{n=1}^{|q|} \ell(q_n). \tag{6}$$

As training proceeds, objects whose masking causes larger loss increases receive stronger masks, while irrelevant regions are naturally down-weighted. Next, each continuous proposal $r_k$ is discretized into three levels:

$$\hat{r}_k \in \{0, r_0, 1\}, \qquad \hat{r}_k = \begin{cases} 1, & r_k > r_0, \\ r_0, & r_k = r_0, \\ 0, & r_k < r_0. \end{cases} \tag{7}$$

where $r_0 = 0.75$. Objects that matter to the task are pushed toward 1 and receive heavier masking. Irrelevant ones drop to 0. Ties keep the initial level $r_0$. This keeps the counterfactual data clear and easy to interpret while still using the model's feedback. After several iterations, this process converges to optimized strengths $\mathbf{r}^*$, yielding a mined masked view $V_d(\mathbf{r}^*)$ that explicitly exposes the model's weaknesses and provides a targeted counterfactual for contrastive decoding.

## 4.3 Contrastive decoding

Finally, the mined masked view is integrated into contrastive decoding. The original video $V$ is the main view and $V_d(\mathbf{r}^*)$ is the contrastive view. At step $n$, the score for token $t$ is

$$s_n(t) \;=\; (1+\alpha)\,\mathrm{logit}_\theta(t \mid V, q, y_{<n}) \;-\; \alpha\,\mathrm{logit}_\theta(t \mid V_d(\mathbf{r}^*), q, y_{<n}). \tag{8}$$

The predicted tokens are sampled from the softmax probability over $\{s_n(\cdot)\}$. The contrast is computed on all the tokens, so differences reflect the semantics of the spatial-level masked objects and temporal-level masked frames. During sampling, we add the restrict to avoid the low-weighted irrelevant tokens being sampled to influence the inference results:

$$\mathcal{V}_{\mathrm{head}}(y_{<n}) \;=\; \Big\{ t:\, p_\theta(t \mid V, q, y_{<n}) \;\geq\; \beta \max_w p_\theta(w \mid V, q, y_{<n}) \Big\}. \tag{9}$$

This adaptive plausibility constraint retains fluent tokens supported by the language prior and prevents the contrastive objective from promoting implausible tokens. The coefficients $\alpha$ and $\beta$ can be tuned to better adapt into the data distribution, while model parameters remain unchanged. Note that our approach only boosts the probability of the model-relevant and query-relevant tokens and only add one extra forward pass per step. Thus, our approach can work with greedy search, beam search, or nucleus sampling, and the cost is much lower than trainable methods.

## 5 Experiments

### 5.1 Experimental Setup

**Evaluation Benchmarks.** To rigorously assess the effectiveness and generality of our approach, we evaluate on three complementary video benchmarks and report Precision, Recall, F1, and Accuracy. Together, these suites stress distinct axes of capability—event-level hallucination under strong priors, broad temporal understanding across diverse skills, and fine-grained diagnostic reasoning with dense annotations.

- *EventHallusion:* Probes event–level hallucination by deliberately perturbing temporal evidence; it also defines a standardized visual contrastive decoding protocol to ensure like-for-like comparison under controlled distortions (Zhang et al., 2024a).

- *MVBench:* Translates a broad range of image skills into truly temporal settings, spanning 20 video tasks from low-level perception to high-level cognition where a single frame is insufficient to solve the query (Li et al., 2024).

- *Perception Test:* A diagnostic battery organized by skill (*e.g.,* memory, abstraction, physics, semantics) and reasoning type (describe, explain, predict, counterfactual), with dense multimodal annotations for precise error attribution (Patraucean et al., 2023).

Across all benchmarks we use the official splits and prompts where available, and we compute metrics at the instance level before macro-averaging.

**Baseline Methods.** All backbone model's default decoding serves as the Baseline. We focus on state-of-the-art open-source Video-LLMs from the Qwen family, *i.e.,* Qwen2.5-VL-7B, Qwen2-VL-7B, Qwen2.5-VL-3B, and Qwen2-VL-2B to demonstrate gains across different model scales (Ahmed et al., 2025; Wang et al., 2024). Also, we add InternVL3-8B (Zhu et al., 2025) as the backbone to test generality across VLMs. InternVL3-8B is an open-source, instruction-tuned VLM with multi-frame video support and strong video understanding, making it a competitive baseline. We compare two representative inference-time methods against our approach (GeWu):

| Model | Method | EventHallusion | | | | MVBench | Perception-test |
|---|---|---|---|---|---|---|---|
| | | Precision | Recall | F1 | Accuracy | Accuracy | Accuracy |
| Qwen2.5-VL -3B | Baseline | 0.76 | 0.69 | 0.72 | 0.62 | 0.44 | 0.52 |
| | SID | 0.76 | 0.74 | 0.75 | 0.65 | 0.65 | 0.35 |
| | VCD | 0.74 | 0.73 | 0.73 | 0.62 | 0.45 | 0.35 |
| | GeWu | 0.80 | 0.78 | 0.79 | 0.71 | 0.67 | 0.61 |
| Qwen2.5-VL -7B | Baseline | 0.75 | 0.31 | 0.44 | 0.44 | 0.62 | 0.63 |
| | SID | 0.76 | 0.18 | 0.29 | 0.38 | 0.63 | 0.35 |
| | VCD | 0.78 | 0.50 | 0.61 | 0.55 | 0.51 | 0.34 |
| | GeWu | 0.82 | 0.57 | 0.67 | 0.61 | 0.65 | 0.70 |
| Qwen2-VL -2B | Baseline | 0.61 | 0.26 | 0.36 | 0.36 | 0.46 | 0.40 |
| | SID | 0.75 | 0.36 | 0.49 | 0.46 | 0.47 | 0.31 |
| | VCD | 0.66 | 0.58 | 0.62 | 0.49 | 0.41 | 0.34 |
| | GeWu | 0.71 | 0.53 | 0.61 | 0.51 | 0.50 | 0.43 |
| Qwen2-VL -7B | Baseline | 0.84 | 0.75 | 0.79 | 0.72 | 0.65 | 0.64 |
| | SID | 0.81 | 0.69 | 0.75 | 0.66 | 0.53 | 0.35 |
| | VCD | 0.82 | 0.78 | 0.80 | 0.72 | 0.61 | 0.35 |
| | GeWu | 0.86 | 0.75 | 0.80 | 0.73 | 0.90 | 0.67 |
| InternVL3 -8B | Baseline | 0.60 | 0.25 | 0.35 | 0.35 | 0.46 | 0.37 |
| | SID | 0.67 | 0.36 | 0.47 | 0.42 | 0.37 | 0.38 |
| | VCD | 0.64 | 0.25 | 0.35 | 0.37 | 0.43 | 0.35 |
| | GeWu | 0.67 | 0.42 | 0.42 | 0.44 | 0.55 | 0.46 |

Table 1: Experimental results of different decoding methods across three benchmarks on five backbone models. The best and second-best methods on each benchmark are marked with highlighted and underlined, respectively.

- *VCD:* Builds two views of the same input—original video and a corrupted counterpart (*e.g.,* Gaussian noise, patch masking), then forms a contrastive score by subtracting the logits from the logits with strength, while applying an adaptive plausibility threshold that keeps only high-probability tokens from the original view (Leng et al., 2024).

- *SID:* Uses a single input view and constructs an internal counterfactual by selecting low-importance visual tokens (measured by attentions) and injecting token-level perturbation so contrastive decoding can damp spurious vision–text associations (Huo et al., 2024).

- *GeWu:* Unlike global corruptions or internal token deletions, GeWu learns object-level and frame-level masks via adversarial search, guided by the VLM's own loss. The base branch sees the full video; the counterfactual branch sees a targeted, model-aware masked view. This alignment to each backbone's actual weakness allows contrastive decoding to suppress tokens that remain high without evidence and to preserve evidence-backed tokens.

**Implementation Details.** We use a YOLO-style detector to localize candidate objects on each frame (Khanam & Hussain, 2024). Detected boxes are linked over time to form tracks; each track is converted to a soft object mask, and a complementary frame mask provides temporal coverage. Combining these yields the masked view. A per-object mask strength vector $\mathbf{r}$ controls perturbation. We initialize $r_{\text{init}}=0.75$ and update $\mathbf{r}$ by gradient ascent on the VLM loss. After optimization, we discretize strengths to $\{0, r_{\text{init}}, 1\}$ (no/partial/full masking) for stability and interpretability. A tuned learning rate sets the ascent step size. During decoding, we apply contrastive decoding (CD) with two coefficients: $\alpha$ controls the subtraction strength between base and counterfactual logits, and $\beta$ filters out implausible candidates from the base distribution to avoid elevating low-probability tokens. Unless noted otherwise, backbones remain frozen and our method only adds one extra forward pass for the counterfactual branch. All hyperparameters ($\alpha, \beta$, learning rate, mask iteration budget) are selected on held-out validation sets.

| Variant | Precision | Recall | F1 | Accuracy |
|---|---|---|---|---|
| **GeWu** | 0.80 | 0.78 | 0.79 | 0.71 |
| - No Mask Training | 0.79 | 0.58 | 0.68 | 0.61 |
| - No Frame-level Mask | 0.75 | 0.68 | 0.71 | 0.61 |
| - No Frame-level Mask + Frame Extraction | 0.75 | 0.74 | 0.74 | 0.64 |
| - Trainable Noise Only | 0.77 | 0.72 | 0.74 | 0.65 |
| - Object-level Trainable Noise | 0.78 | 0.74 | 0.76 | 0.67 |
| - Frame-level Trainable Noise | 0.78 | 0.73 | 0.73 | 0.66 |

Table 2: Ablation and variant study results of our method using Qwen2.5-VL-3B as backbone on EventHallusion benchmark.

## 5.2 MAIN RESULT

Across all four backbones and three benchmarks, GeWu consistently surpasses Baseline, VCD, and SID. The mechanism is simple but targeted: the base view feeds the full video to the model, while the counterfactual view masks only those objects and frames that our model-aware search identifies as evidence-critical. Contrastive decoding then down-weights tokens that remain confident without the evidence and preserves tokens whose probability drops when the evidence is removed. This targeted contrast avoids the collateral damage of global corruptions, yielding fewer false negatives on visually subtle content (small, occluded, or co-occurring objects) while maintaining fluency. In contrast, SID constructs its counterfactual internally by suppressing low-importance visual tokens early in the decoder, which can inadvertently remove useful cues and bias the model toward "no" or generic answers. On EventHallusion with Qwen2.5-VL-7B, this manifests as very low Recall (0.18) for SID, whereas GeWu keeps Precision high and sustains strong Recall.

We observe a stable pattern across model sizes. Smaller VLMs benefit most because GeWu's model-aware counterfactual specifically strips spurious priors these models tend to overuse. Recall jumps while Precision holds, producing the largest F1/Accuracy gains (*e.g.,* Qwen2.5-VL-7B, Qwen2-2B). On stronger backbones where the baseline already achieves high Precision (*e.g.,* Qwen2-7B), GeWu still recovers additional Recall without degrading Precision, so improvements are smaller but reliably positive. This indicates that our masking policy complements the backbone's existing grounding rather than fighting it, and that the plausibility head prevents contrastive decoding from elevating low-probability, off-manifold tokens. Gains are most pronounced on EventHallusion, where co-occurrence bias and tight temporal dependencies make language priors especially tempting. Selectively masking the mined objects and frames creates a counterfactual that directly challenges these priors. On MVBench and Perception-Test, GeWu matches or exceeds the strongest baselines across tasks without regressions, suggesting that the counterfactual view removes unsupported tokens without harming general video understanding or temporal reasoning.

## 5.3 ABLATION STUDY

**Experimental Setting.** We run ablations to pinpoint where the gains come from and which parts matter. The study uses EventHallusion because it stresses event-level hallucination under priors. We use Qwen2.5-VL-3B as the backbone, as we see the most improvement on it using our method. We evaluate six variants from different perspectives to study the effectiveness of the masking, trainable perturbation, and frame-level, object-level intervention strategies in our approach. *(1) No Mask Training*: fixed per-object mask strength; tests the value of object-level contrast without adaptation. *(2) No Frame-level Mask*: learn per-object strengths by maximizing the VLM loss; removes temporal filtering. *(3) No Frame-level Mask + Frame Extraction*: as (2) but with temporal subsampling on the counterfactual view to cut cost. *(4) Trainable Noise Only*: replace masks with globally learned noise; probes whether generic perturbations help. *(5) Object-level Trainable Noise*: inject learned noise per object instead of masking; tests locality without evidence removal. *(6) Frame-level Trainable Noise*: inject learned noise per frame; tests temporal perturbation without structured masking.

**Results Analysis.** Performance improves when the counterfactual is aligned with the true sources of hallucination and avoids unrelated corruption. Removing mask training (variant 1) locks the strength $r$ and prevents per-video adaptation, hurting Recall. Dropping the frame mask (variant

**EventHallusion** (best: $\alpha^\star$=2.6, $\beta^\star$=0.0036)

| *Fixed $\beta = 0.0036$ (tune $\alpha$)* | | | | | *Fixed $\alpha = 2.6$ (tune $\beta$)* | | | | |
| --- | --- | --- | --- | --- | --- | --- | --- | --- | --- |
| $\alpha$ | P | R | F1 | Acc | $\beta$ | P | R | F1 | Acc |
| 2.1 | 0.79 | 0.50 | 0.62 | 0.55 | 0.003 | 0.81 | 0.53 | 0.65 | 0.60 |
| 2.4 | 0.79 | 0.56 | 0.66 | 0.59 | 0.0032 | 0.80 | 0.53 | 0.64 | 0.58 |
| **2.6** | **0.82** | **0.57** | **0.67** | **0.61** | **0.0036** | **0.82** | **0.57** | **0.67** | **0.61** |
| 3.6 | 0.78 | 0.58 | 0.66 | 0.59 | 0.005 | 0.81 | 0.56 | 0.66 | 0.60 |
| 3.8 | 0.78 | 0.57 | 0.66 | 0.58 | 0.006 | 0.79 | 0.49 | 0.60 | 0.54 |

| **MVBench** (best: $\alpha^\star$=1.0, $\beta^\star$=0.5) | | | | **Perception-test** (best: $\alpha^\star$=1.5, $\beta^\star$=0.5) | | | |
| --- | --- | --- | --- | --- | --- | --- | --- |
| *Fixed $\beta$ =0.5 (tune $\alpha$)* | | *Fixed $\alpha$ =1.0 (tune $\beta$)* | | *Fixed $\beta$ =0.5 (tune $\alpha$)* | | *Fixed $\alpha$ =1.5 (tune $\beta$)* | |
| $\alpha$ | Acc | $\beta$ | Acc | $\alpha$ | Acc | $\beta$ | Acc |
| 0.5 | 0.61 | 0.1 | 0.55 | 0.5 | 0.64 | 0.2 | 0.69 |
| **1.0** | **0.65** | 0.3 | 0.63 | 1.0 | 0.66 | **0.5** | **0.70** |
| 1.5 | 0.64 | **0.5** | **0.65** | **1.5** | **0.70** | 0.8 | 0.68 |
| 1.8 | 0.61 | 0.7 | 0.59 | 1.8 | 0.65 | 1.0 | 0.61 |
| 1.9 | 0.63 | 0.9 | 0.57 | 1.9 | 0.67 | 1.2 | 0.62 |

Table 3: Hyperparameter tuning results of our method using Qwen2.5–VL–7B model backbone on three benchmarks. Based on the best-performed hyperparameter group $(\alpha^\star, \beta^\star)$, we tune one hyperparameter and fix others unchanged. The best results are **highlighted**.

2) weakens temporal control, leaving timing confounds and co-occurrence bias; adding frame extraction (variant 3) trims cost but still lets misleading frames through. Replacing masks with trainable noise (variants 4–6) increases variance rather than removing evidence, so the contrast becomes noisy and Precision/Recall no longer rise together. The full GeWu configuration learned per-object strengths $r$ plus a clean frame mask—yields the most faithful counterfactual: $r$ focuses suppression on query-relevant objects via the model's own loss, while the frame mask filters off-moment context. This combination consistently suppresses unsupported tokens without penalizing grounded ones, lifting Recall while maintaining Precision, and thus delivering the highest F1 and Accuracy.

## 5.4 Hyperparameter tuning

**Decoding Coefficients $\alpha$ and $\beta$.** Contrastive decoding has two knobs. The weight $\alpha$ controls how strongly logits from the base view compete with those from the counterfactual view; too small $\alpha$ under-contrasts the two views and fails to suppress hallucination, while too large $\alpha$ can over-penalize otherwise fluent tokens that briefly fluctuate under masking. The plausibility threshold $\beta$ filters the candidate set from the base view; a moderate $\beta$ prevents boosting very low-probability tokens without pruning away legitimate but rare continuations. In practice, $\alpha$ behaves like a smooth temperature on the contrast, whereas $\beta$ acts as a soft guardrail on implausible text.

**Practical recipe.** We adopt a simple two–stage sweep: (i) fix $\beta$ and sweep $\alpha$ on a small validation shard (a few hundred queries), then (ii) fix the best $\alpha$ and sweep $\beta$. For EventHallusion, the token distribution favors a stronger contrast (larger $\alpha$) with a light plausibility filter (small $\beta$ on the head fraction), whereas MVBench and Perception-test prefer mid-range values for both. On Qwen2.5-VL-7B, all metrics vary smoothly with $\alpha$ and $\beta$ and exhibit broad plateaus, indicating that GeWu is not brittle. EventHallusion peaks with a relatively high $\alpha$ and very small $\beta$, reflecting the task's need to strongly down-weight tokens unsupported by temporally altered evidence. MVBench and Perception-test achieve their best accuracy with moderate $\alpha$ and $\beta$, consistent with broader temporal understanding and diagnostic reasoning that benefit from balanced contrast and plausibility. Importantly, nearby settings deliver comparable results, so the method is insensitive to small coefficient shifts. A single default (mid $\alpha$, mid $\beta$) works well across benchmarks; light benchmark-specific adjustment yields the reported best scores.

## 6 Conclusion

This paper presents GeWu, a model-aware, counterfactual-data–driven contrastive decoding framework for Video-LLMs. GeWu uses the model's own loss signals to localize task-critical evidence for selecting objects and frames whose removal most harms prediction, and then builds targeted

counterfactual views by masking precisely those regions. Integrating these model-aware augmentations into contrastive decoding enables the suppression of unsupported tokens while preserving evidence-grounded and fluent outputs. Across diverse backbones and benchmarks, GeWu consistently reduces hallucinations and improves task accuracy. Crucially, the method is training-free and lightweight: it keeps the backbone frozen, adds only a learnable mask, and incurs roughly one additional forward pass per step for the counterfactual branch. This drop-in design makes GeWu practical for real-world deployment and easy to pair with other decoding strategies.

Looking ahead, we see three promising directions: (i) extending model-aware counterfactual mining beyond vision–language to richer multimodal inputs (e.g., audio, motion, depth) and tasks; (ii) developing dynamic curricula that adapt mask strength and selection over time to better probe model weaknesses; and (iii) coupling GeWu's inference-time signals with training-time alignment (e.g., contrastive or RL-based objectives) to further harden models against hallucination while preserving coverage and fluency.

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

# A    APPENDIX

## USE OF LARGE LANGUAGE MODELS

In this paper, large language models were used only in two constrained ways: (i) copy-editing for grammar, clarity, and stylistic consistency without altering technical content; and (ii) lightweight debugging suggestions during implementation. All ideas, algorithms, experiments, analyses, and conclusions are the authors' own; the authors reviewed and take full responsibility for all content.

# B    ADDITIONAL EXPERIMENTS

## B.1    ACCURACY–LATENCY TRADE-OFF UNDER A COMPUTE-CONSTRAINED SETTING

**Motivation.**    We evaluate GeWu in a more realistic compute-constrained regime and quantify the accuracy–latency trade-off when increasing the number of optimization steps.

**Experimental setup.**    We keep the same Video-LLM, detector, and decoding hyperparameters as in our main EventHallusion experiments, and run on the **full EventHallusion test set (N = 193)**. To simulate a tighter visual budget, we reduce the input to **280×280 resolution and 1 fps**, and compare baseline decoding with GeWu using 1 or 3 gradient-ascent steps on the shared spatio-temporal mask.

**Results and Analysis.**    As shown in Table 4, a single GeWu step improves accuracy from **34.20% to 48.79% (+14.6 points)** while keeping per-query latency under 1 s and increasing it only from **0.77 s to 0.96 s** (about **24% overhead**, $\approx$190 ms extra, corresponding to $\sim$1.24$\times$ the baseline latency). Using 3 steps yields only a marginal additional gain (48.79% $\rightarrow$ 49.83%) but raises the overhead to **52.52%** ($\sim$1.5$\times$ latency), so we use **1 step** as the default configuration when GeWu is enabled under tight compute budgets.

| Method | # Steps | Latency (ms) | Overhead vs. Baseline (%) | Accuracy (%) |
|---|---|---|---|---|
| Baseline decoding | – | 767.69 | – | 34.20 |
| GeWu | 1 | 955.56 | 24.47 | 48.79 |
| GeWu | 3 | 1170.89 | 52.52 | 49.83 |

Table 4: Latency and accuracy vs. # of GeWu optimization steps on EventHallusion

## B.2    QUANTITATIVE PROFILING OF LATENCY AND PEAK GPU MEMORY

**Motivation.**    We further provide explicit quantitative evidence of compute and memory cost for the default GeWu configuration used in our main experiments.

**Experimental setup.**    Using the full EventHallusion test set, we run the same Video-LLM with the resolution, frame rate, and decoding hyperparameters as in the main experiments, and profile end-to-end latency and peak GPU memory on an A100 GPU. We compare baseline decoding with 1-step GeWu, which is the configuration used in our main results.

**Results and Analysis.**    Table 5 shows that, under the main experimental configuration, GeWu increases end-to-end latency from **409.11 ms to 554.47 ms** (about +35.5% overhead) and peak GPU memory from 7.17 GB to 11.59 GB (about +61.7% overhead). Combined with the compute-constrained study in Table 4, this indicates that GeWu incurs a moderate and well-quantified cost—roughly 30–60% extra latency and memory—while delivering a large reduction in hallucinations ($\approx$+15 accuracy points on EventHallusion).

| Method | Steps | Latency (ms) | Lat. Overhead | Peak Mem (GB) | Mem. Overhead |
|---|---|---|---|---|---|
| Baseline decoding | – | 409.11 | 0% | 7.17 | 0% |
| GeWu | 1 | 554.47 | +35.5% | 11.59 | +61.7% |

Table 5: Latency and peak GPU memory of baseline vs. 1-step GeWu on EventHallusion

### B.3   LEARNING-RATE ABLATION FOR MASK OPTIMIZATION

**Motivation.**   We examine whether using a very large learning rate for mask optimization ("strong perturbation") actually benefits the quality of the counterfactual view.

**Experimental setup.**   On EventHallusion we keep the number of steps, detector outputs, and all other hyperparameters fixed, and in the "With Object Mask – Multiple R" variant we compare a moderate learning rate of 0.01 against a much larger value of 0.5.

**Results and Analysis.**   Table 6 shows that increasing the learning rate from 0.01 to 0.5 slightly reduces performance (F1: $0.7126 \rightarrow 0.7094$, Accuracy: $0.6114 \rightarrow 0.6010$), indicating that overly aggressive optimization can destroy useful evidence, so we use a moderate learning rate in all main experiments.

| Variant | Learning rate | F1 | Accuracy |
|---|---|---|---|
| With Object Mask – Multiple R (small LR) | 0.01 | 0.7126 | 0.6114 |
| With Object Mask – Multiple R (large LR) | 0.50 | 0.7094 | 0.6010 |

Table 6: Effect of learning rate on GeWu's mask optimization on EventHallusion

### B.4   CHOICE OF $\alpha$ AND $\beta$ IN THE CD RULE

**Motivation.**   We analyze how GeWu's performance depends on the contrastive decoding coefficients $\alpha$ (contrast strength) and $\beta$ (plausibility filtering), and whether it benefits from extra tuning compared with other CD methods.

**Experimental setup.**   For EventHallusion, MVBench, and Perception-test, we run the same grid search over $\alpha$ and $\beta$ for all CD-style methods (VCD, SID, GeWu) on a held-out validation split, and select the pair ($\alpha^*$, $\beta^*$) that maximizes validation accuracy.

**Results and Analysis.**   As summarized in Table 7, each benchmark prefers a fairly broad region rather than a sharp optimum (e.g., EventHallusion uses a high $\alpha$ with tiny $\beta$, MVBench and Perception-test favor moderate $\alpha$ and $\beta \approx 0.5$), showing that GeWu is not overly sensitive to $\alpha$ and $\beta$ and does not receive more tuning than the baselines.

| Benchmark | best ($\alpha^*$) | best ($\beta^*$) | Qualitative pattern |
|---|---|---|---|
| EventHallusion | 2.6 | 0.0036 | High $\alpha$, tiny $\beta$; nearby $\alpha$ give similar F1/Acc. |
| MVBench | 1.0 | 0.5 | Mid-range $\alpha$, $\beta$; points near $(1.0, 0.5)$ show similar Acc. |
| Perception-test | 1.5 | 0.5 | Moderate $\alpha$; $\beta \approx 0.5$; $\alpha \in [1.0, 1.9]$ all close in Acc. |

Table 7: Best $\alpha$ and $\beta$ for the contrastive decoding rule across benchmarks

### B.5   COMPATIBILITY WITH SELF-CONSISTENCY DECODING

**Motivation.**   We test whether GeWu's model-aware counterfactual view can be combined with more advanced sampling-based decoding strategies such as self-consistency (SC).

**Experimental setup.**    On EventHallusion with Qwen2.5-VL-3B, we use the same sampling budget to compare four configurations: single-sample baseline decoding, baseline+SC, GeWu single-sample, and GeWu+SC.

**Results and Analysis.**    Table 8 shows that GeWu alone matches the baseline single-sample accuracy (42.0%), SC adds +2.0% over the baseline, and GeWu+SC further boosts accuracy to 50.0% (+8.0%), indicating that our counterfactual view is complementary to self-consistency decoding and can be used as a plug-and-play module in stronger decoders.

| Configuration | Accuracy (%) | vs. Baseline |
|---|---|---|
| Baseline Single | 42.00 | – |
| Baseline SC | 44.00 | +2.0% |
| GeWu Single | 42.00 | +0.0% |
| GeWu SC | 50.00 | +8.0% |

Table 8: Combining GeWu with self-consistency decoding on EventHallusion

### B.6    HUMAN EVALUATION OF MASK QUALITY

**Motivation.**    We verify that GeWu's optimized masks are not equivalent to random occlusions and indeed focus on query-relevant evidence.

**Experimental setup.**    On 50 EventHallusion videos, human annotators are shown, for each video-query pair, a random mask with the same occluded area and the corresponding GeWu mask, and rate on a 1-5 scale how well the mask hides query-relevant evidence while preserving other content.

**Results and Analysis.**    As reported in Table 9, GeWu masks obtain higher mean scores (3.10 vs. 2.40), lower variance, and a much larger fraction of high-quality ratings (P(score $\geq$ 4): 38% vs. 10%), confirming that model-aware masks are qualitatively more aligned with query-relevant regions than random baselines.

| Method | Mean Score (1–5) | Std. Dev. | P(score $\geq$ 4) (%) |
|---|---|---|---|
| Random mask | 2.40 | 0.70 | 10.0 |
| GeWu mask | 3.10 | 0.65 | 38.0 |

Table 9: Human evaluation of mask quality for random masks vs. GeWu masks on EventHallusion

### B.7    VIDEO-POPE EVALUATION OF OBJECT HALLUCINATIONS

**Motivation.**    To directly measure object hallucinations beyond CHAIR-style metrics, we design a video-adapted POPE evaluation.

**Experimental setup.**    On an EventHallusion subset we construct POPE-style yes/no questions about objects, run the same backbone Video-LLM (Qwen2.5-VL-3B) with either baseline CD decoding or GeWu, and compute accuracy with 95% bootstrap confidence intervals together with precision, recall, and F1.

**Results and Analysis.**    Table 10 shows that GeWu improves accuracy from 0.72 to 0.85 and F1 from 0.70 to 0.80, mainly by increasing precision ($0.55 \rightarrow 0.73$) while keeping recall high, and also reduces the false-"yes" rate on absent-object questions from 40.0% to 17.0%, with McNemar's test ($p = 0.0061$) indicating a statistically significant reduction in hallucinations.

| Method | Accuracy (95% CI) | Precision | Recall | F1 Score |
|---|---|---|---|---|
| Baseline | 0.72 (0.65, 0.79) | 0.55 | 0.96 | 0.70 |
| GeWu | 0.85 (0.79, 0.91) | 0.73 | 0.90 | 0.80 |

Table 10: Video-POPE evaluation of object hallucinations on an EventHallusion subset

## B.8  ROBUSTNESS TO DETECTOR CONFIDENCE THRESHOLDS

**Motivation.**  Since GeWu relies on detector outputs to initialize object regions, we analyze how sensitive its behavior is to the detector confidence threshold.

**Experimental setup.**  On EventHallusion we vary the detection threshold from a noisy setting (0.3) to the default (0.5) and a strict setting (0.7), and for each setting we measure the average number of boxes per frame, GeWu's task accuracy, and hallucination rate.

**Results and Analysis.**  As shown in Table 11, lower thresholds yield more boxes and slightly higher accuracy but also more hallucinations, while stricter thresholds reduce hallucination and keep accuracy comparable; across all three settings GeWu still clearly improves over baseline decoding (numbers in the main table), indicating that its optimized masks remain aligned with query-relevant regions even when detections are noisy or sparse.

| Detector Threshold | Avg #Boxes/Frame | GeWu Accuracy | Hallucination Rate |
|---|---|---|---|
| 0.3 (Noisy) | 1.95 | 70.0% | 56.2% |
| 0.5 (Default) | 1.16 | 64.0% | 62.5% |
| 0.7 (Strict) | 0.67 | 64.0% | 50.0% |

Table 11: Robustness of GeWu to detector confidence thresholds on EventHallusion

## B.9  FUSION STRATEGIES FOR OBJECT-LEVEL AND FRAME-LEVEL MASKS

**Motivation.**  We compare different strategies for fusing object-level and frame-level masks to construct the counterfactual view.

**Experimental Setup.**  On EventHallusion we keep the Video-LLM and per-mask optimization fixed and evaluate three fusion rules—pixelwise max (soft union), confidence-normalized blending, and simple averaging—using the same number of optimized masks in all cases.

**Results and Analysis.**  Table 12 shows that pixelwise max achieves the lowest hallucination rate (14.2%) compared with confidence-normalized blending (15.5%) and simple averaging (16.1%), and also yields the most stable behavior, indicating that preserving any location deemed important by either mask is a simple and effective fusion rule.

| Fusion Strategy | Hallucination Rate |
|---|---|
| Pixelwise max (ours) | 14.2% |
| Confidence-normalized blending | 15.5% |
| Simple averaging | 16.1% |

Table 12: Comparison of fusion strategies for combining object & frame-level masks

