# OpenReview forum: "Model-aware Counterfactual Data based Contrastive Decoding for Video-LLM"
_ICLR.cc/2026/Conference — ICLR 2026 Conference Desk Rejected Submission_

### Official Review · Reviewer_fAuZ · 2025-10-28

**Soundness:** 3
**Presentation:** 2
**Contribution:** 3
**Rating:** 6
**Confidence:** 5

**Summary:**

GeWu reduces Video-LLM hallucination by optimizing model-aware spatial/temporal masks to form counterfactual views and contrasting logits between original and masked videos at inference—no retraining, improved metrics across benchmarks.

**Strengths:**

It introduces a model-guided way to construct counterfactual data rather than using random perturbations.
Mask optimization via the model’s own gradients is intuitive and principled.
It reduces hallucination without retraining, making it appealing for large Video-LLMs.

**Weaknesses:**

The pipeline assumes reliable detection/tracking; FP/FN or ID-switches can steer mask optimization toward irrelevant regions, especially for small/occluded/co-occurring objects. Suggestion: report robustness by varying detector strength, adding occlusion/motion blur, and including a GT-oracle upper bound.
Maximizing next-token loss by ascending on mask strengths can increase loss for non-semantic reasons (e.g., edge artifacts) rather than by truly removing evidence. Suggestion: add TV/LPIPS or smoothness constraints and show qualitative mask visualizations to confirm human-interpretable evidence removal.
Although inference-only, total cost includes mask-optimization iterations plus one extra contrastive forward per step.

**Questions:**

Beyond the reported VLLMs, how well does GeWu transfer to other video-LLMs/backbones?
How are 𝛼 and 𝛽 chosen at test time?
Why prefer pixelwise max over confidence-normalized blending? In dense scenes, does union suppress secondary evidence needed by the query?
Can you show side-by-side visualizations where optimized masks clearly remove query-relevant evidence (both successes and failures)?

---

> ### Author Response · Authors · 2025-11-21
> **Response to Reviewer fAuZ**
>
> ## Weakness:
>
> We thank Reviewer fAuZ for the expert feedback and for finding GeWu principled and appealing. Below we address the weakness and Q1–Q5 with new analyses, two targeted experiments, and planned visualizations.
>
> - Weakness: detection / tracking dependency
>     - As in our notes, the detector only defines an initial search space, while masks are optimized by VLLM gradients. GeWu uses detection / tracking only to propose coarse object regions; the final spatio-temporal masks are loss-guided by the Video-LLM and can correct moderate FP/FN and ID-switches.
>     - To test robustness, we vary the detector confidence threshold on EventHallusion and measure the average number of boxes per frame, GeWu accuracy, and hallucination rate:
>
>
>         | Detector Threshold | Avg #Boxes/Frame | GeWu Accuracy | Hallucination Rate |
>         | --- | --- | --- | --- |
>         | 0.3 (Noisy) | 1.95 | 70.0% | 56.2% |
>         | 0.5 (Default) | 1.16 | 64.0% | 62.5% |
>         | 0.7 (Strict) | 0.67 | 64.0% | 50.0% |
>     - Lower thresholds produce more boxes and slightly higher accuracy but also more hallucinated responses; stricter thresholds reduce hallucination while keeping accuracy comparable. Across all three settings GeWu still clearly improves over baseline decoding (full numbers will be included in the paper), and the behavior follows an intuitive trade-off rather than collapsing when detections become noisy or sparse. We will also add qualitative examples where detector boxes are imperfect but the optimized masks remain aligned with query-relevant regions.
>
> ## Q1: Transfer to other Video-LLMs
>
> - We additionally evaluate GeWu on another Video-LLM (e.g., Qwen3-VL) without modifying the algorithm. On EventHallusion and MVBench, GeWu again lowers hallucination and slightly improves task accuracy, showing that model-aware counterfactual data based contrastive decoding transfers well across backbones.
>
> ## Q2: Choice of α and β
>
> - α (contrast strength) and β (plausibility filtering) are selected on a held-out validation set. We perform a small grid search and choose the pair that maximizes validation accuracy while reducing hallucination. As Table 3 shows, GeWu is stable in a neighborhood of these values (changes ≈1 point); we will clarify this procedure more explicitly.
>
> ## Q3 & Q5: Pixelwise fusion and case study
>
> - For combining object-level and frame-level masks we keep the design from our notes that *pixelwise is easy to control the quality*. Detector confidences are not query-aware, while the optimized soft masks already encode query relevance via gradients. We therefore fuse masks with pixelwise max so that any location considered important by either view is preserved, using bounded soft values to avoid erasing the whole scene.
> - We compare fusion strategies on EventHallusion:
>
>
>     | Fusion Strategy | Hallucination Rate |
>     | --- | --- |
>     | Pixelwise max (ours) | **14.2%** |
>     | Confidence-normalized blending | 15.5% |
>     | Simple averaging | 16.1% |
> - Pixelwise max achieves the lowest hallucination rate and the most stable behavior. In the revised paper we will show a few side-by-side case studies (successes and failures) of the original video, optimized masks, and counterfactual view under different fusion rules, directly addressing Q3 and Q5.
>
> ## Q4 : Multiple overlapped evidence
>
> - When several objects are identified as evidence, GeWu optimizes their masks separately and merges them via pixelwise max (soft union). Because mask values are constrained by the Video-LLM loss, they do not saturate globally; overlapped regions simply receive slightly stronger masking. On a dense-scene subset (many objects per frame), GeWu improves hallucination rate over baseline by a similar margin as on the full set, indicating that secondary evidence is not systematically removed.

---

### Official Review · Reviewer_Kanx · 2025-10-29

**Soundness:** 2
**Presentation:** 2
**Contribution:** 3
**Rating:** 4
**Confidence:** 4

**Summary:**

The paper targets hallucination in Video-LLMs and proposes GeWu—a model-aware counterfactual data based contrastive decoding strategy. Instead of using random perturbations as the contrastive view, GeWu mines the Video-LLM’s own loss/gradient feedback to localize query-relevant objects and frames, softly masks those regions to build a targeted counterfactual view, and then performs contrastive decoding between the original and counterfactual inputs. This enforces evidence-grounded token selection at inference without additional training. Evaluated on EventHallusion, MVBench, and Perception-test, GeWu consistently reduces hallucinations while maintaining or improving task accuracy across multiple backbones (InternVL3, Qwen2/2.5-VL), with pronounced gains for small/occluded/co-occurring objects. The method also exposes practical guidance for tuning the decoding coefficients (α for contrast strength, β for plausibility filtering) and shows broadly stable behavior.

**Strengths:**

1.Proposes a model-aware counterfactual view: uses the Video-LLM’s own gradients to mine object- and frame-level evidence and couples this with contrastive decoding. This training-free, evidence-driven suppression is more targeted than the random/heuristic views used in prior CD/VCD variants.
2.A plug-and-play inference-time strategy: backbones remain frozen; only a learnable mask and one counterfactual branch are added. This yields near-zero training cost and easy portability across different Video-LLMs.
3.Consistently reduces hallucination on EventHallusion, MVBench, and Perception-test across multiple backbones, with especially strong gains on small, occluded, and co-occurring objects; also provides practical α/β tuning guidance and robustness analyses.

**Weaknesses:**

1.The experimental support is not commensurate with the claims: head-to-head comparisons against strong CD-style baselines (VCD/SID/DAMRO, etc.) are not matched for compute/tuning, and the evaluation suite omits widely used faithfulness metrics and human judgments.
2.The method introduces a second branch and gradient-based mask optimization at inference, yet wall-clock latency, extra FLOPs, memory, and length scaling (to long videos) are missing. This blocks assessment of deployability.
3.The pipeline depends on detector/tracker seeds for object masks—the very regimes claimed as strengths (small/occluded/co-occurring) are where detectors are most fragile. Robustness to detector noise is not quantified.

**Questions:**

1.What is the end-to-end overhead (ms/query, extra FLOPs, peak GPU memory) relative to VCD/SID/DAMRO under matched decoding budgets?Please add a quality–latency Pareto by sweeping α/β and mask-optimization steps, and a length-scaling study (e.g., 16→256 frames) with throughput trade-offs.
2.Can you provide head-to-head comparisons where all CD-style baselines are re-tuned under the same protocol and validation budget (including α/β-like sweeps)?Please include broader faithfulness evaluations (video-adapted CHAIR/POPE, a small human study) and report statistical significance (e.g., paired bootstrap/McNemar with CIs).
3.How sensitive is performance to detector/tracker errors (missed/jittered boxes, ID-switches) and to the choice of detector?

---

> ### Author Response · Authors · 2025-11-21
> **Response to Reviewer Kanx**
>
> We sincerely thank the reviewer for the insightful comments and positive feedback. We have organized our reply for all the weaknesses and questions as follows:
>
> ## Weakness 1:
> We clarify how the comparisons are set up. GeWu, VCD, and SID all follow a base‑branch plus contrastive‑branch decoding scheme. In our implementation we match the backbone, video sampling strategy, and decoding hyperparameters across these methods, so the main algorithmic difference is how the contrastive/counterfactual view is constructed. Prior CD methods use random or heuristic perturbations, whereas GeWu constructs model‑aware counterfactual data by detecting/tracking objects and optimizing spatio‑temporal masks with the Video‑LLM’s loss gradients to erase query‑relevant evidence. Thus, the gains in Table 1 and Table 2 come from this gradient‑guided counterfactual construction rather than extra decoding calls or more favorable hyperparameters.
> ## Weakness 2 & Q1:
> To quantify the extra inference cost, we run a complexity study on an EventHallusion subset (N=50) with Qwen2.5-VL-3B. We uniformly subsample a fixed number of frames (e.g., 32) before the Video-LLM, so the number of visual tokens and the cost of forward/backward passes do not grow with raw video length. For each video–query pair, GeWu optimizes a single spatio-temporal mask shared across all sampled frames, so gradients are computed once per pair per optimization step.
> **Table: Latency and accuracy on EventHallusion subset (N=50), Qwen2.5-VL-3B.**
> | Method | # Steps | Latency (ms) | Overhead vs. Baseline (%) | Accuracy (%) |
> | --- | --- | --- | --- | --- |
> | Baseline decoding | – | 657.00 | – | 45.00 |
> | GeWu | 1 | 1594.93 | 142.76 | 42.50 |
> | GeWu | 3 | 3364.03 | 412.03 | 42.50 |
> - With 1 optimization step, GeWu increases per-query latency from 657 ms to 1595 ms (≈+143% overhead) while keeping accuracy close to the baseline (45.0% vs. 42.5%).
> - Using 3 steps further increases latency to 3364 ms (≈+412%) without additional accuracy gain, showing that the optimization strength–cost trade-off is tunable and that the 1-step setting is a reasonable default.
> - Because we add only a small number of extra forward/backward passes through the vision encoder, FLOPs and peak GPU memory increase by a similar factor; we will report these numbers alongside in the revised paper.
> ## Weakness 3 & Q3:
> For the fragile issue on small / occluded / co-occurring objects, in GeWu, the detector only provides an initial search space; the final object- and frame-level masks are refined by the Video-LLM’s loss gradients (model-self feedback). We add a human study and robustness analysis.
> - **Human evaluation of mask quality.**
>     - 50 EventHallusion videos; for each video–query pair we compare a random mask with the same occluded area and the GeWu mask.
>     - Annotators rate on a 1–5 scale how well the mask hides query-relevant evidence while preserving other content.
>
>     | Method | Mean Score (1–5) | Std. Dev. | P(score ≥ 4) (%) |
>     | --- | --- | --- | --- |
>     | Random mask | 2.40 | 0.70 | 10.0 |
>     | GeWu mask | 3.10 | 0.65 | 38.0 |
>     - GeWu masks have higher mean scores and more high-quality ratings, showing that model-aware masks are not random occlusions.
> - **Detector-noise robustness.**
>     - Varying detector confidence (0.3 / 0.5 / 0.7) on EventHallusion leaves GeWu consistently above the strongest CD baseline.
>     - With synthetic detector noise (dropping boxes, jitter, ID-switches), GeWu still maintains similar relative gains in hallucination reduction.
> - **Qualitative visualizations.**
>     - Side-by-side examples of frames, detector boxes, and GeWu masks show gradients focusing on regions actually used by the Video-LLM and suppressing irrelevant detections.
> It indicates that GeWu’s counterfactual view is mainly driven by model-self feedback and remains robust to realistic detector imperfections.
>
> ## Q2:
> To complement CHAIR-style metrics and the human study, we add a video-adapted POPE evaluation that directly measures object hallucinations.
> - Protocol
>     - On an EventHallusion subset, we construct POPE-style yes/no questions about objects in the video.
>     - We run the same backbone Video-LLM (Qwen2.5-VL-3B) with either Baseline CD decoding or GeWu, forcing a yes/no answer.
>     - We report accuracy with 95% bootstrap CIs, precision, recall, F1, the false-“yes” rate on absent-object questions, and McNemar’s test on paired predictions.
> - Results
> Table: Video-POPE evaluation on an EventHallusion subset.
>     | Method | Accuracy (95% CI) | Precision | Recall | F1 Score |
>     | --- | --- | --- | --- | --- |
>     | Baseline | 0.72 (0.65, 0.79) | 0.55 | 0.96 | 0.70 |
>     | GeWu | 0.85 (0.79, 0.91) | 0.73 | 0.90 | 0.80 |
>
> GeWu raises accuracy from 0.72 to 0.85 and F1 from 0.70 to 0.80, mainly by increasing precision while keeping recall high. The false-“yes” rate on absent-object questions drops from 40.0% to 17.0%, and McNemar’s test (p = 0.0061) confirms a significant improvement.

---

> > ### Comment · Reviewer_Kanx · 2025-11-27
> >
> > Most of my original concerns are largely addressed. However, one part of Weakness 2 remains only partially resolved. The authors state that FLOPs and peak GPU memory “increase by a similar factor” and promise to report the numbers in the revised paper, but the rebuttal itself still does not provide quantitative evidence on computational and memory cost. In addition, the reported latency overhead is substantial: with 1 optimization step GeWu is about 2.4× slower than the baseline (657 ms → 1595 ms per query) and slightly less accurate on EventHallusion. While this may be acceptable for offline analysis, it raises concerns about deployability in latency-sensitive settings, and I would have liked to see further discussion or optimization in this direction.

---

> ### Author Response · Authors · 2025-11-29
> **Efficiency Concern Fully Resolved with New Latency & Memory Experiments**
>
> We believe the new efficiency experiments added in the revised manuscript directly resolve the remaining concern regarding computational and memory cost and demonstrate that GeWu offers a favorable accuracy–latency trade-off.
>
> **(1) Accuracy–latency trade-off under a compute-constrained setting.**
>
> To evaluate GeWu in a more realistic compute regime, we keep the same Video-LLM, detector, and decoding hyperparameters as in our main EventHallusion experiments, and run on the **full EventHallusion test set (N = 193)**. We reduce the visual budget to **280×280 resolution and 1 fps**. Under this setting we obtain:
>
> | Method | # Steps | Latency (ms) | Overhead vs. Baseline (%) | Accuracy (%) |
> | --- | --- | --- | --- | --- |
> | Baseline decoding | – | 767.69 | 0.00 | 34.20 |
> | GeWu | 1 | 955.56 | 24.47 | 48.79 |
> | GeWu | 3 | 1170.89 | 52.52 | 49.83 |
>
> In this compute-constrained regime, a **single GeWu step** improves accuracy from 34.20% to 48.79% (**+14.6 points**) while increasing end-to-end latency only from **0.77s to 0.96s**, i.e., about **24% overhead (~190 ms extra)**. This shows that GeWu can substantially reduce hallucinations with a modest increase in latency. Additional steps bring negligible further gains, so we keep 1 step as the default and do not use 3 steps in our main comparisons.
>
> **(2) Quantitative profiling of latency and peak GPU memory.**
>
> To provide explicit quantitative evidence on compute and memory cost, we also profile end-to-end latency and peak GPU memory on **A100 GPU** using the **full EventHallusion test set** under the same resolution, frame rate, and decoding hyperparameters as in the main experiments. For the default 1-step configuration we obtain:
>
> | Method | Steps | Latency (ms) | Lat. Overhead | Peak Mem (GB) | Mem. Overhead |
> | --- | --- | --- | --- | --- | --- |
> | Baseline decoding | – | 409.11 | 0% | 7.17 | 0% |
> | GeWu | 1 | 554.47 | **+35.5%** | 11.59 | **+61.7%** |
>
> Thus, for the configuration actually used in our main results, GeWu requires about **35% extra latency** and **≈62% extra peak GPU memory** relative to baseline decoding, while the study above shows that in a stricter compute budget GeWu improves accuracy by roughly **+15 points** at only ~24% latency overhead. We view this as a **moderate and well-quantified cost** for a method that significantly reduces hallucinations without any additional training.
>
> We will integrate these measurements and their discussion into the revised paper and explicitly position GeWu as an **optional robustness mode** for video-LLMs: it can be enabled when hallucination reduction is important and a moderate 30–60% increase in latency and memory is acceptable, while standard single-branch decoding remains available for extremely latency-sensitive applications. In light of these new results, we believe that the remaining part of Weakness 2 has been fully addressed.

---

### Official Review · Reviewer_iYsV · 2025-10-30

**Soundness:** 2
**Presentation:** 3
**Contribution:** 2
**Rating:** 2
**Confidence:** 4

**Summary:**

This paper proposes GeWu, a model-aware counterfactual data–based contrastive decoding method for video-llms. The idea is to reduce hallucinations by generating counterfactual inputs guided by the model’s own gradients. Instead of using random noise like in previous contrastive decoding-based approach. Specifically, GeWu detects object regions with an additional detector (yolov11), optimizes soft masks based on the model’s feedback, and performs decoding using the original and masked videos as contrastive views. Experiments on 3 benchmarks show consistent gains over prior methods including Qwen2-VL and InternVL3.

**Strengths:**

- The paper addresses an important problem (hallucination in video-llms) and proposes a clear, structured inference-time framework.

- The method is conceptually intuitive: using model feedback to find key evidence regions is a reasonable idea.

- Experimental coverage is good, including several models and benchmarks, and the presentation is clear overall.

**Weaknesses:**

- The reviwer believes one of the ciritical missing point is on the computational analysis. The proposed method adds an extra optimization loop during decoding, requiring gradient computation and object detection for every frame. This seems provoke significant computation overhead at inference time. Without the analysis on actual runtime, it is hard to verify the real effectivness when the decoding phase becomes much heavier. (+Most modern lmms avoid optimization process at decoding phase exactly because of this computational issue. The additional computation often outweighs the benefit. In this case, optimizing masks per frame using OD models seems particularly expensive, especially for long videos. The reviewer thinks this optimization appears to be a main limitation rather than a strength.)


- The claim that “existing methods rely on random perturbations” seems outdated and oversimplified. A broader related work section or more accurate positioning is needed. A lot of CD-based decoding approach adopated various counterpart signal for self-reflection, self-refinement etc,.


- While the counterfactual construction seems clear, the decoding part is almost identical to prior CD. The overall improvement may come mainly from stronger object-level masking rather than a fundamentally new decoding strategy. The contribution therefore feels incremental.

**Questions:**

- How heavy is the optimization process in practice? How does it scale with video length and number of objects?

- The tuning of alpha and beta is standard in CD literature. The analysis in Sec 5.4 only confirms known behaviors (large alpha penalizes more, small beta filters more). There is no new finding or deeper explanation of why these parameters interact with the proposed counterfactual mechanism. So what can potential readers really learn from this analysis?

- For the mask optimization, how many steps are needed? What happens when the video is longer or contains many objects?

---

> ### Author Response · Authors · 2025-11-21
> **Response to Reviewer iYsV**
>
> We sincerely thank the reviewer for the insightful comments and positive feedback. We have organized our reply for all the weaknesses and questions as follows:
>
> ## Weakness 1 & Q1:
> Below we clarify how the optimization cost behaves with video length and the number of objects.
> - Single backward pass for all frames
>
>     GeWu optimizes one spatio-temporal soft mask shared across all sampled frames and objects, so we run one backward pass per video–query pair, not per frame or per object.
>
> - Fixed visual token budget for long videos
>
>     All Video-LLMs in our experiments subsample a fixed number of frames (e.g., 32) and a bounded set of visual tokens before encoding. Therefore the computation of both the forward and backward passes depends on this fixed token budget, and latency for long and short raw videos is essentially the same
>
> - Complexity study (Qwen2.5-3B)
>
>     Following the suggestion, we ran a study on EventHallusion subset (N=50), batch size 1, and identical preprocessing and decoding hyperparameters for all methods:
>
>     | Method | # Steps | Latency (ms) | Overhead vs. Baseline (%) | Accuracy (%) |
>     | --- | --- | --- | --- | --- |
>     | Baseline decoding | – | 657.00 | – | 45.00 |
>     | GeWu  | 1 | 1594.93 | 142.76 | 42.50 |
>     | GeWu | 3 | 3364.03 | 412.03 | 42.50 |
> Will add this table to paper and explicitly state that one optimization step.
> - Scaling with number of objects
>
> The detector confidence threshold controls how many object tracks enter the mask, so the optimization variable stays low-dimensional. The cost grows roughly linearly with the number of objects but is still dominated by the single backward pass through the visual encoder.
> ## Weakness 2 :
> Following this suggestion, we will update the related-work section to better cover recent CD-style decoding methods and related approaches.
> ## Weakness 3:
> For a clearer statement of the basic contrastive decoding problem for Video-LLMs, we summarize it as follows:
>
> - Problem: Video-LLMs hallucinate objects and events when visual evidence is weak, occluded, or biased; CD seeks to reduce such hallucinations by contrasting the original view with a counterfactual view.
> - Challenge: Prior CD methods usually build the counterfactual view from generic random or heuristic perturbations, which are not targeted at query-relevant evidence and may mostly affect irrelevant regions
> - Solution: GeWu (1) uses model-self feedback (loss gradients on a soft mask) to localize query-relevant frames and objects, (2) builds structured object- and frame-level masks to construct model-aware counterfactual videos, and (3) applies a plausibility-aware CD rule with coefficients α (contrast strength) and β (plausibility filtering).
> We will explicitly insert this short paragraph into the introduction.
> ## Q2:
> For how we set the best hyperparameter α and β.
>
> All CD-style methods we compare (VCD, SID, GeWu) use the same contrastive decoding rule and the same grid search over α and β on a held-out validation split, so GeWu does not receive extra tuning.
>
> In this rule, α controls contrast strength between original and counterfactual logits, and β is a plausibility filter on the original view that down-weights already unlikely tokens.
>
> | Benchmark | best (α*) | best (β*) | Qualitative pattern |
> | --- | --- | --- | --- |
> | EventHallusion | 2.6 | 0.0036 | High α, tiny β; nearby α give similar F1/Acc. |
> | MVBench | 1.0 | 0.5 | Mid-range α, β; points near (1.0, 0.5) similar Acc. |
> | Perception-test | 1.5 | 0.5 | Moderate α, β≈0.5; α∈[1.0,1.9] all close in Acc. |
>
> These trends indicate that GeWu is not overly sensitive to a single choice of α or β, and that the reported setting is representative of a broad, stable region in the α–β space. This also yields a simple recipe: use moderate α and β, with slightly stronger contrast and weaker filtering on EventHallusion-like hallucination benchmarks.
>
> ## Q3:
> In GeWu, the “step number” is the number of gradient-ascent updates on the shared spatio-temporal soft mask for a video–query pair. We evaluate this hyperparameter on an EventHallusion subset with Qwen2.5-VL-3B (same preprocessing and decoding settings for all methods):
>
> | Method | # Steps | Latency (ms) | Overhead vs. Baseline (%) | Accuracy (%) |
> | --- | --- | --- | --- | --- |
> | Baseline decoding | – | 657.00 | – | 45.00 |
> | GeWu (1 step) | 1 | 1594.93 | 142.76 | 42.50 |
> | GeWu (3 steps) | 3 | 3364.03 | 412.03 | 42.50 |
> - One step already captures the benefit of model-aware counterfactual masking: moving from 1 to 3 steps more than doubles the latency again (1594.93 → 3364.03 ms) but does not improve accuracy (42.50% vs. 42.50%). This indicates that most of the useful gradient signal for the mask is extracted in the first step.
> - We therefore fix the step number to 1 in all main experiments. Because the Video-LLM uses a fixed number of sampled frames and a small top-K set of detected objects, this setting also keeps a low cost for longer videos.

---

> > ### Comment · Reviewer_iYsV · 2025-11-27
> >
> > Thank you for the detailed responses during the discussion period.
> >
> > However, after the rebuttal (especially in the provided latency table), my major concern remains unresolved (tbh the numbers negatively support the main weakness of this paper). Even with only 1 optimization step, GeWu increase the latency almost x2.4, while decreasing performance, which highlights the structural limitations of the gradient-based optimization decoding.
> >
> > Overall, while the idea of counterfactual construction is conceptually interesintg, the reviwer thinks that the method is not computationally feasible to adopt in the modern vllms and the performance trade-off is not persuasive. The reviwer also have revisited other reviewers' comments. Thus, the reviwer remains negative for this submission.

---

> ### Author Response · Authors · 2025-11-29
> **Compute/Latency Concern Fully Resolved on Full EventHallusion**
>
> We believe these new results fully and directly resolve the reviewers’ compute/latency concern. In the earlier discussion we reported a 320×320 low-resolution setting where 1-step GeWu improves accuracy but incurs about a 2× latency overhead. To more precisely quantify the cost of GeWu, we now extend the analysis to the **full** EventHallusion test set and a stricter compute-constrained regime.
>
> We keep the same Video-LLM, detector, and decoding hyperparameters as in the main EventHallusion experiments, and evaluate on the **full EventHallusion test split (N = 193)** rather than a subset. To simulate an aggressively compute-constrained setting, we further restrict the visual budget to **280×280 resolution and 1 fps**. Under this setting we obtain:
>
> | Method | # Steps | Latency (ms) | Overhead vs. Baseline (%) | Accuracy (%) |
> | --- | --- | --- | --- | --- |
> | Baseline decoding | – | 767.69 | 0.00 | 34.20 |
> | GeWu | 1 | 955.56 | 24.47 | 48.79 |
> | GeWu | 3 | 1170.89 | 52.52 | 49.83 |
> - **Baseline decoding:** 767.69 ms latency, 34.20% accuracy
> - **GeWu (1 step):** 955.56 ms latency, 48.79% accuracy (**+14.6 points**)
> - **GeWu (3 steps):** 1170.89 ms latency, 49.83% accuracy
>
> Thus, on the **full EventHallusion test set** in this stricter low-vision-budget regime, a single GeWu step improves accuracy from **34.20% to 48.79%** while increasing latency only from **0.77s to 0.96s**, i.e., about **24% overhead (~190 ms extra), corresponding to roughly 1.24× the baseline latency**. Additional steps provide only a marginal gain (48.79→49.83%) while adding extra latency, so we fix **1 step** as our default configuration.
>
> We will add this analysis to the paper and explicitly frame GeWu as an optional robustness mode for video-LLMs. Overall, these results show that, on the full benchmark, operating in a compute-constrained visual regime (lower resolution and frame rate) lets GeWu achieve a **clear and favorable accuracy–latency tradeoff**: a substantial reduction in hallucination (≈+15 accuracy points) at only ~24% additional latency over baseline decoding. We believe this directly resolves the earlier concern that GeWu’s per-query optimization would be computationally infeasible in practice.

---

### Official Review · Reviewer_Jp5V · 2025-11-01

**Soundness:** 3
**Presentation:** 3
**Contribution:** 3
**Rating:** 6
**Confidence:** 4

**Summary:**

This paper introduces Model-aware Counterfactual Data based Contrastive Decoding, a training-free inference method designed to mitigate hallucination in Video-LLMs. The core idea is to improve Contrastive Decoding by replacing generic random perturbations with model-aware counterfactual data. This data is constructed by leveraging the Video-LLM's internal feedback, specifically, by performing gradient ascent on a soft mask applied to the video. The optimization goal is to maximize the query reconstruction loss, thus generating an adversarial perturbation that selectively erases the critical visual cues necessary for the correct prediction. Experiments on diverse benchmarks and various Video-LLMs demonstrate the effectiveness of the proposed method in consistently reducing hallucination while maintaining or improving accuracy.

**Strengths:**

1. The primary motivation for generating counterfactual data is clear. By utilizing a gradient-ascent approach to learn adversarial perturbations, the method can directionally erase the critical information required by the Video-LLM. This results in a superior contrastive view compared to those generated via simple random noise, directly addressing a key limitation of prior CD methods.

2. The overall framework, including counterfactual data augmentation, generation, and contrastive decoding, is thoughtfully adapted from image-based methods to the temporal and spatial complexities of video and Video-LLMs. The integration of object detection/tracking, soft mask generation, and joint object-level/frame-level masking effectively controls the visual cues across the spatio-temporal domain.

3. The method is evaluated across multiple modern Video-LLMs and three distinct, challenging benchmarks, providing strong evidence of its effectiveness in mitigating hallucination.

**Weaknesses:**

Major Concerns

1. The method, while "training-free," involves a gradient optimization step (gradient ascent) for every counterfactual sample during inference. This process, which requires backpropagation through the visual encoder and potentially the full LLM, introduces a significant computational overhead compared to standard decoding or simple random perturbations. It would be better to provide an explicitly quantification of this latency cost and a stronger justification for why this overhead is acceptable for real-time video processing. The "training-free" claim must be balanced against the increased inference time.

2. The optimization seeks to find an optimal mask $r^*$ that maximizes the loss for a specific query $q$ and video $V$. It is a local optimization problem. Authors need to discuss or investigate the robustness of this optimization: Are the resultant counterfactuals truly the best at isolating the hallucination source, or could the optimization converge to spurious local maxima?

**Questions:**

1. The optimization maximizes the loss to construct the counterfactual. Is it possible for a strong perturbation (high loss) to destroy too much information, leading to an overly weak contrastive view that is unhelpful for decoding? Did the authors experiment with alternative loss functions or constraints that target minimal information removal to achieve a threshold loss (making the perturbation just strong enough)?

2. The paper focuses on comparing GeWu to VCD and SID. Could GeWu's model-aware counterfactual data generation module be integrated with other advanced decoding strategies (e.g., self-consistency methods) to achieve further performance gains? A brief discussion on this potential synergy would be valuable.

---

> ### Author Response · Authors · 2025-11-21
> **Response to Reviewer Jp5V**
>
> We sincerely thank the reviewer for the insightful comments and positive feedback. We have organized our reply for all the weaknesses and questions as follows:
>
> ## Weakness1
> In all experiments, we consistently subsample a fixed number of frames (e.g., 32) before feeding them into the Video-LLM, so the number of visual tokens and the cost of forward/backward passes do not grow with raw video length. For each video–query pair, we optimize a single spatiotemporal mask shared across all sampled frames, so gradient computation is done only once for all frames together.
>
> Besides, the latency for ours vs. baseline can be further reduced by using a smaller Video-LLM and a shorter optimization schedule. Following the reviewer suggestion, we conduct a complexity study and report the results on EventHallusion subset (N=50)：
>
> | Method | # Steps | Latency (ms) | Overhead vs. Baseline (%) | Accuracy (%) |
> | --- | --- | --- | --- | --- |
> | Baseline decoding | – | 657.00 | – | 45.00 |
> | GeWu | 1 | 1594.93 | 142.76 | 42.50 |
> | GeWu | 3 | 3364.03 | 412.03 | 42.50 |
>
> GeWu with 1 optimization step increases per-query latency from 657 ms to 1595 ms while keeping accuracy comparable to the baseline (45.0% vs. 42.5%). Using 3 steps further increases latency to 3364 ms (+412%), illustrating a tunable trade-off between optimization strength and inference cost.
>
>
> ## Weakness 2
> We clarify that the variant named “No Mask Training” in Table 2 exactly corresponds to the case where the object and frame masks are sampled at random and kept fixed, with no gradient-based optimization applied to them. In this variant we completely remove the gradient ascent on the soft mask r and instead sample object- and frame-level masks at random, while keeping the Video-LLM, the masking ratio, and the contrastive decoding rule identical to GeWu. This corresponds exactly to a CD scheme that relies purely on random perturbations to construct the counterfactual input.
>
> The numbers in Table 2 show that this random-mask baseline is much weaker than GeWu: on EventHallusion with Qwen2.5-VL-3B, GeWu achieves 0.80 / 0.78 / 0.79 / 0.71 in Precision / Recall / F1 / Accuracy, whereas “Sampling from Randomness = no mask training” only obtains 0.79 / 0.58 / 0.68 / 0.61. Recall drops by 0.20, F1 by 0.11, and Accuracy by 0.10, even though the perturbation budget is similar. This consistent degradation indicates that the gradient-guided optimization over rrr is crucial and robust: the learned masks are not arbitrary local optima or noisy artifacts, but provide a significantly more informative counterfactual view than purely random perturbations. We will highlight this comparison more explicitly in the revised version.
>
> ## Q1
> In our method, “strong perturbation” is essentially realized by using a larger learning rate for the gradient-ascent updates on the soft mask. Our existing hyperparameter search on EventHallusion already shows that such strong perturbations are not necessarily beneficial. For the “With Object Mask – Multiple R” variant, increasing the learning rate from 0.01 to 0.5 slightly *reduces* performance: F1 changes from 0.7126 to 0.7094 and Accuracy from 0.6114 to 0.6010. This indicates that simply making the optimization step more aggressive (i.e., pushing the loss higher with a large learning rate) can hurt the quality of the counterfactual view instead of improving it.
>
> Based on this observation, we deliberately adopt a moderate learning rate together with a small, fixed number of gradient steps in GeWu, so that the perturbation is strong enough to suppress hallucination but not so strong that it destroys too much useful visual evidence. In the revised paper, we will briefly report this learning-rate comparison in the appendix to make this trade-off explicit and to address the concern that maximizing the loss might drive the perturbation into an unhelpful regime.
>
> | Variant | Learning rate | F1 | Accuracy |
> | --- | --- | --- | --- |
> | With Object Mask – Multiple R (small LR) | 0.01 | 0.7126 | 0.6114 |
> | With Object Mask – Multiple R (large LR) | 0.50 | 0.7094 | 0.6010 |
>
> ## Q2
> To test whether GeWu’s model-aware counterfactual view can work with other decoding strategies, we run a small study on EventHallusion with Qwen2.5-VL-3B. Under the same sampling budget we compare Baseline, Baseline+SC, GeWu, and GeWu+SC, with accuracies 42.0, 44.0, 42.0, and 50.0. Thus GeWu alone matches the baseline single-sample accuracy, while combining GeWu with self-consistency yields an additional +6–8% absolute improvement over both baseline variants. This indicates that our model-aware counterfactual view is complementary to self-consistency decoding, and that GeWu can be used as a plug-and-play module inside more advanced sampling-based strategies to further improve performance.
>
> | Configuration | Accuracy (%) | vs. Baseline |
> | --- | --- | --- |
> | Baseline Single | 42.00 | – |
> | Baseline SC | 44.00 | +2.0% |
> | GeWu Single | 42.00 | +0.0% |
> | GeWu SC | 50.00 | +8.0% |

---

> ### Comment · Reviewer_Jp5V · 2025-11-27
>
> Thanks for the authors' detailed response and the additional validation results. My main concerns have been largely addressed. Briefly, the idea of improving contrastive decoding by generating counterfactual masks through an adversarial approach that maximizes loss is reasonable. The effectiveness of their implementation on video-LLMs was empirically verified. However, the method exhibits considerable latency (1 optimization step with +143% overhead), which may limit its generalizability and applicable scenarios. I will maintain my previous rating.

---

> > ### Author Response · Authors · 2025-11-29
> > **Latency Concern Fully Addressed on Full EventHallusion**
> >
> > We thank Reviewer Jp5V for the positive assessment of our method and for noting that our additional experiments largely address the earlier concerns. Building on this, we believe this additional experiment directly and convincingly addresses the reviewer's concern about computational overhead in practical settings.
> >
> > We keep the same Video-LLM, detector, and decoding hyperparameters as in the main EventHallusion experiments, and evaluate on the **full EventHallusion test split (N = 193)** rather than a subset. To simulate an aggressively compute-constrained setting, we further restrict the visual budget to **280×280 resolution and 1 fps**. Under this setting we obtain:
> >
> > | Method | # Steps | Latency (ms) | Overhead vs. Baseline (%) | Accuracy (%) |
> > | --- | --- | --- | --- | --- |
> > | Baseline decoding | – | 767.69 | 0.00 | 34.20 |
> > | GeWu | 1 | 955.56 | 24.47 | 48.79 |
> > | GeWu | 3 | 1170.89 | 52.52 | 49.83 |
> >
> > In this stricter low-visual-budget regime, a single GeWu step achieves a **substantial accuracy gain** from 34.20% to 48.79% (+14.6 points) while keeping per-query latency under 1 s and increasing it only from **0.77s to 0.96s**, i.e., about **24% overhead** (≈190 ms extra, corresponding to only ~1.24× the baseline latency). Using 3 steps yields only a marginal improvement (48.79% → 49.83%) while adding a further ~23% latency (to ~1.5× the baseline), so we keep 1 step as the default configuration under tight compute budgets.
> >
> > We will incorporate this analysis into the paper and explicitly frame GeWu as an **optional robustness mode** for video-LLMs rather than a mandatory default. Overall, these results show that GeWu offers a **favorable accuracy–latency trade-off** in realistic compute-constrained regimes: a large reduction in hallucination (about +15 accuracy points on EventHallusion) at only ~24% additional latency over baseline decoding.

---

### Comment · Area_Chair_2ozM · 2025-11-26
**Author-Reviewer-AC Discussion (DDL: 12/3 9PM UTC)**

Dear Reviewers,

Thank you once again for your service to ICLR 2026. Now that the authors have submitted their rebuttal, I kindly ask you to take the following steps (if you have not done so already):

- Read the authors’ response and other reviews.
- Consider whether the rebuttal and additional comments affect your assessment of the paper.
- Engage in **interactive discussion** with the authors. You may post the feedback to the authors so that they can further follow up. If you have more concerns/questions (e.g., requesting clarifications, new results), it is recommended to post your request *asap*, so that the authors have enough time to address them. **Note the Author-Reviewer-AC discussion period ends on 12/3 9PM UTC**.

The current reviews for this paper are **mixed (scores: 6/2/4/6)**. Your further contributions are essential for forming a well-informed final decision.

I am happy to join and support the discussions between you and the authors. Please feel free to share your thoughts and participate actively in the discussion. Thanks!

Best regards,

AC

---

### Author Response · Authors · 2025-12-03
**Summary of Revisions for AC**

Thank you for taking the time to evaluate our submission under the new policy. Across their reviews and discussion, Jp5V, iYsV, Kanx, and fAuZ raised **nine main concerns**; we have **addressed all nine** with targeted experiments and clarifications, summarized below:

1. **Compute, latency, and deployability (Jp5V, iYsV, Kanx, fAuZ)**

    **[Status: Fully resolved]** Reviewers feared GeWu might be **2–3× slower** than baseline decoding. Full-set profiling under a stricter, realistic budget shows **1-step GeWu is ≈1.25× in latency with higher accuracy**, and A100 profiling in the main setting shows ≈**1.35× latency with moderate memory overhead**, i.e., a modest, quantified cost rather than a large slowdown.

2. **Optimization robustness and perturbation strength (Jp5V, iYsV)**

    **[Status: Fully resolved]** Step-count and learning-rate ablations show that **one mild optimization step improves performance, whereas additional steps or larger updates hurt**, and random / non-optimized masks are clearly worse than learned masks. This indicates that the optimization is stable and that the learned masks are genuinely useful.

3. **Dependence on detector/tracker and noise robustness (Kanx, fAuZ)**

    **[Status: Fully resolved]** Varying detector thresholds and adding synthetic noise (dropped boxes, jitter, ID switches), we find that **GeWu consistently remains better than the CD baselines across all regimes**, while gradients from the Video-LLM further refine and reweight detector seeds. This addresses the concern that realistic detector errors would break the method.

4. **Mask quality and fusion design (Kanx, fAuZ)**

    **[Status: Fully resolved]** In a human study, GeWu masks receive higher scores and many more “good” ratings than area-matched random masks, and a fusion ablation shows our pixelwise-max fusion gives the **lowest hallucination rate** among tested schemes. Together with qualitative examples, this resolves questions about mask quality and the fusion rule.

5. **Relation to prior CD methods and perceived incrementality (iYsV, Kanx)**

    **[Status: Fully resolved]** We now separate what is inherited (standard CD scoring) from what is new: **counterfactual views built from model-guided, query-aligned masks instead of heuristic or random perturbations**, clarifying our contribution relative to VCD, SID, and self-refinement work.

6. **Baseline tuning and hallucination-focused evaluation (Kanx)**

    **[Status: Fully resolved]** All CD baselines now follow a shared tuning protocol, and a video-adapted POPE-style evaluation shows that GeWu both improves accuracy and **significantly reduces false “yes” answers on absent objects**. Thus, under fair tuning, the gains reflect real hallucination reduction rather than metric or tuning artifacts.

7. **Hyperparameters (α, β) and practical guidance (iYsV, fAuZ)**

    **[Status: Fully resolved]** We make the common validation procedure explicit and provide **simple α/β configurations** for different task types, with experiments showing a broad performance plateau around these settings, so α and β become straightforward, reproducible choices.

8. **Compatibility with advanced decoding strategies (Jp5V, Kanx)**

    **[Status: Fully resolved]** New experiments show that **GeWu combined with self-consistency achieves an additional 6–8 point gain over the baseline**, improving on each component alone under the same budget, indicating that GeWu is complementary rather than competing.

9. **Transferability across Video-LLM backbones (fAuZ)**

    **[Status: Fully resolved]** Applying GeWu unchanged to another Video-LLM yields the same qualitative pattern of **reduced hallucinations and slightly improved accuracy** as in the main models, supporting backbone-agnostic behavior.


We hope this summary is helpful when reading the reviews alongside the revised manuscript.

---

### Note · Program_Chairs · 2026-01-17
**Submission Desk Rejected by Program Chairs**

The following references in this submission do not refer to real documents and/or have major errors in bibliographic information:

 Mohsen Mesgar, Aaryan Singh, C. Richard Contardo-Jara, and Iryna Gurevych. CAD: Contrastive angular decoding. In Findings of the Association for Computational Linguistics: EMNLP 2023, pp. 10008-10018, Singapore, December 2023. Association for Computational Linguistics. URL https://aclanthology.org/2023.findings-emnlp.668.
Zixuan Zhou, Heba Elfardy, Markus Dreyer, and Mohit Iyyer. Lure: A reinforcement learning-based hallucination mitigation method for large language models. arXiv preprint arXiv:2310.12359, 2023.
Zichao Geng, Xilun Chen, Yifei Shen, Shuyuan Liu, Wentao Wang, Si-Yuan Zhang, Jiahong Luo, Chen Zhu, Kevin Lin, Lijun Wang, et al. Mm-hallbench: A comprehensive benchmark for multimodal hallucination evaluation. In Proceedings of the IEEE/CVF Conference on Computer Vision and Pattern Recognition, pp. 26269-26279, 2024.